# Lipid Metabolism Heterogeneity and Crosstalk with Mitochondria Functions Drive Breast Cancer Progression and Drug Resistance

**DOI:** 10.3390/cancers14246267

**Published:** 2022-12-19

**Authors:** Aurelien Azam, Nor Eddine Sounni

**Affiliations:** 1Cancer Metabolism and Tumor Microenvironment Group, GIGA-Cancer, University of Liège, 4000 Liège, Belgium; 2Laboratory of Tumor and Development Biology, GIGA-Cancer, University of Liège, 4000 Liège, Belgium

**Keywords:** lipid metabolism, breast cancer, heterogeneity, hypoxia, tumor microenvironment, drug resistance, oxidative stress, cancer progression

## Abstract

**Simple Summary:**

Metabolic heterogeneity and mitochondrial function are important parameters that might influence therapeutic vulnerabilities and predict the clinical outcomes of cancer patients. While mutations in oncogenes and tumor suppressors can stimulate cell-autonomous metabolic reprogramming, it became clear that the effects of tumor microenvironment (interactions with extracellular matrix and stromal cells and tumor hypoxia) impose a selective pressure on the metabolic preferences of cancer cells. Here, we review recent reports about lipid metabolism heterogeneity in breast cancer progression and therapy response. We describe processes intrinsic to breast cancer cells and extrinsic factors inflicted by the tumor microenvironment that regulate lipid metabolism pathways and breast cancer resistance.

**Abstract:**

Breast cancer (BC) is a heterogeneous disease that can be triggered by genetic alterations in mammary epithelial cells, leading to diverse disease outcomes in individual patients. The metabolic heterogeneity of BC enhances its ability to adapt to changes in the tumor microenvironment and metabolic stress, but unfavorably affects the patient’s therapy response, prognosis and clinical effect. Extrinsic factors from the tumor microenvironment and the intrinsic parameters of cancer cells influence their mitochondrial functions, which consequently alter their lipid metabolism and their ability to proliferate, migrate and survive in a harsh environment. The balanced interplay between mitochondria and fatty acid synthesis or fatty acid oxidation has been attributed to a combination of environmental factors and to the genetic makeup, oncogenic signaling and activities of different transcription factors. Hence, understanding the mechanisms underlying lipid metabolic heterogeneity and alterations in BC is gaining interest as a major target for drug resistance. Here we review the major recent reports on lipid metabolism heterogeneity and bring to light knowledge on the functional contribution of diverse lipid metabolic pathways to breast tumorigenesis and therapy resistance.

## 1. Introduction

Breast Cancer (BC) is a heterogenous and complex disease that emerges from epithelial cells lining the milk ducts and milk-secreting cells. It includes a large panel of molecular subtypes defined by the expression of the estrogen receptor (ER), progesterone receptor (PR) and human epidermal growth factor 2 receptor (HER2) [1,2]. The use of common molecular subtypes of BC such as luminal A (ER^+^/PR^+^/HER2^−^), luminal B (ER^+^/PR^−^/HER2), HER2-positive (ER^−^/PR^−^/HER2^+^) and triple negative breast cancer (TNBC, ER^−^/PR^−^/HER2) has advanced patient management and standard treatments [3]. However, each group has also diverse gene expression profiles and molecular diversities that can be found within individual tumors and within inter-individual patients (Figure 1) [4]. For instance, the expression of HER2 may vary between primary tumors and their metastatic cells, which causes challenges for treatment efficacy and biomarker characterization [5]. The molecular heterogeneity of BC can be attributed to genetic, epigenetic and non-genetic alterations in tumor epithelial cells and cells within the tumor microenvironment (TME) as well. Transformations can occur in multipotent mammary stem cells as well as in more differentiate cells [6], which can be followed by a malignant conversion switch, leading to metastasis to the lymph nodes or hematological metastasis to lung, brain and bone [7]. Distinct molecular profiles can impact the phenotype and the diverse biological behavior of breast cancer cells within primary tumors and metastasis [8]. Besides, malignant cells evolve within a heterogeneous TME that contains serval sub-populations of cancer-associated fibroblasts (CAFs) and infiltrating immune cells in a complex extracellular matrix (ECM). Cancer cells can shape ECM, induce immunosuppression [9] and gain metabolic plasticity during malignant progression. These capabilities are also translated by metabolic changes that support adaptation to nutrient scare and metabolic stress in a harsh TME [10]. Due to their high proliferative rates, BC cells increase their intake of nutrients and activate the anabolic pathway of lipid synthesis to sustain tumor growth. This review will explore factors that affect lipid metabolic heterogeneity and its consequences on BC cancer progression and therapy response.

## 2. Plasticity of Lipid Metabolism

Tumor cells evolve in a dynamic environment that influences their metabolic diversity and that can be explained by their plasticity and fineness. The metabolic adaptation of growing tumors has been attributed to metabolic change to aerobic glycolysis [11], known as Warburg effect, while change in lipid metabolism has gained interest in metastasis and therapy resistance [12]. Indeed, the role of lipid metabolism in metastasis and drug resistance became recently evident in many cancer types [13,14,15,16]. Thus, tracking lipid metabolism heterogeneity might help uncover vulnerabilities that could be new targets for BC resistance. Several factors specific to cancer and stroma cells may dictate lipid metabolic heterogeneity in tumors. Hence, cell-intrinsic and cell-extrinsic factors are essential for shaping tumor lipid metabolism at various levels and in multiple cellular compartments [10].

### 2.1. Fatty Acid Uptake, Storage and De Novo Lipogenesis in Cancer

Lipids are the main structural components of cell membrane compartment that also play a vital role as energy sources and signaling molecules. Dysregulated lipid metabolism pathways in breast cancer have received renewed interest, as oncogenic signaling and molecular heterogeneity regulate lipidomics and the progression of BC [17]. The key metabolic hallmarks of BC cells are alterations in fatty acid (FA) transport, uptake, de novo lipogenesis, storage and oxidation to generate ATP (adenosine triphosphate) (Figure 2) [12].

#### 2.1.1. Lipid Uptake

There are two ways for mammalian cells to obtain lipids: extracellular uptake and de novo lipogenesis (also called de novo FA synthesis). Lipid uptake is mostly common in normal cells; the bloodstream provides lipids ingested in food as free FA or low-density lipoproteins (LDL), while the de novo lipogenesis is restricted to specialized cells such as adipocytes and hepatocytes [18,19]. Extracellular uptake is facilitated by membrane-associated transport proteins, including fatty acid-transport protein-1 (FATP1), the scavenger receptor fatty acid-translocase (FAT, or CD36) and fatty acid-binding proteins (FABPs). The transcription factor sterol regulatory element-binding protein 1 (SREBP1) regulates the expression of these enzymes and plays a central role in lipogenesis [20]. Although FA can diffuse across phospholipid (PL) bilayers, much of the FA uptake is enabled by the increased expression of integral or membrane-associated proteins in cancer cells (FATP1 and CD36) or the expression of FABP4 in the TME by adipocytes and endothelial cells [13]. In ovarian cancer, FABP4 has a key role in lipid transport from omental tissue [21]. In models of therapy-induced hypoxia, FABP4 expression in the TME increases the formation of lipid droplets (LD) in cancer cells and contributes to cancer resistance to oxidative stress and ferroptosis [13,22]. In BC, circulating FABPs derived from adipocytes in obese women have been associated with cancer progression and contributed to the progression of the multistage mammary tumor of the MMTV-TGF-α mouse, a preclinical model that recapitulates major steps of human BC development [23]. Once in the cytosol, FA are bound by cytoplasmic FABPs before entering metabolic or signaling pathways [24]. To be used in metabolic pathways, free FA must be activated by conversion to fatty acyl-coenzyme A (CoA) by long-chain acyl-coenzyme A synthetases (ACSLs), enter mitochondria through carnitine palmitoyltransferase 1 (CPT1) and undergo fatty acid oxidation (FAO). More extracellular sources of FA can be used by cancer cells using the endo-lysosome, which processes LDL into FA, and via macropinocytosis. Cancer cells can also obtain FA from intracellular sources such as LD, PL and from de novo lipogenesis [25].

#### 2.1.2. Lipid Storage

Lipid droplets (LD) provide FA actively through lipolysis and the lipophagy of triacylglycerols (TAG), mainly to fuel the mitochondrial oxidative metabolism during nutrient deprivation [26]. They consist of lipid storage organelles containing a neutral lipid core, mainly composed of triglycerides (TGA) and cholesteryl esters (CE), and surrounded by a phospholipid (PL) monolayer [27]). Lipid droplets maintain lipid homeostasis, prevent lipotoxicity and generate ATP by breaking down lipids stored in LD during conditions of metabolic stress [24]. Lipid droplets are dynamically synthetized via the de novo synthesis of TAG and CE within the lipid bilayers of the endoplasmic reticulum (ER) [28]. Briefly, excess lipids within the ER intermembrane are released into the cytoplasm to form LD. To provide TAG, LD might be degraded by lipolysis, a reaction catalyzed by adipose triacylglycerol lipase (ATGL), hormone-sensitive lipase (HSL) and monoacylglycerol lipase (MAGL) [29]. An excess of lipolysis may lead to lipotoxicity, characterized by a harmful increase in cytoplasmic-free FA (FFA) and mitochondrial FAO, leading to the production of ROS. This phenomenon is regulated by Perilipin 5, which inhibits ATGL-mediated lipolysis [30]. Lipid droplets can also be degraded by lipophagy, a form of autophagy in which LD are incorporated into autophagosomal membranes, fused with lysosomes and hydrolyzed [12,31]. Lipid droplets are reported to play a key role in cancer adaptation to therapy-induced hypoxia [16,32] and drive cancer progression through epithelial mesenchymal transition in the acidic TME [33]. Yet, the molecular regulation of LD lipophagy remains unknown. Fatty acids can be also provided by the hydrolysis of PL. They are deacylated by phospholipases A and B (PLAs and PLBs) to produce lysophospholipids (LPL), which can then be further deacylated via lysophospholipase A (LPLAs) to produce glycerophosphate and a FFA [25].

#### 2.1.3. Lipogenesis

During de novo lipogenesis, cells use excess carbohydrates such as glucose, or use amino acids (glutamine) and acetate uptake from extracellular acetate to form new FA. This process directly links lipid metabolism to glucose and glutamine metabolism and is catalyzed by three main enzymes named ATP-citrate lyase (ACLY), acetyl-CoA carboxylase (ACC) and the multifunctional fatty acid synthase (FASN). The expression of these enzymes is regulated by SREBP1 and mTOR signaling in breast cancer [34]. Citrate produced in the TCA cycle can be derived from glycolysis or from the reductive carboxylation of glutamine in cancer cells under hypoxia [35,36]. Then, citrate is liberated into the cytosol and converted into acetyl-CoA (Ac-CoA) by ACLY [37], which is then carboxylated to malonyl-CoA by ACC in an ATP-dependent manner. Finally, FASN converts seven malonyl-CoA and one priming Ac-CoA into a 16-carbon saturated (16:0) FA, palmitate (or palmitic acid (PA)), the initial product of FA synthesis [24]. Further complex FA might be generated from PA via elongation by elongases (ELVOLs) [38] and via saturation and conversion into monounsaturated fatty acids (MUFA) of various lengths and degrees of saturation by stearoyl-CoA desaturases (SCDs) and fatty acid desaturases (FADs) [39]. Fatty acids may be incorporated into diacylglycerols (DAG), TAG, or converted into PL, such as phosphatidic acid (PA), phosphatidylethanolamine (PE) and phosphatidylserine (PS) [24,40]. In order to be stored in cells, FA, under the form of DAG, are finally converted into TG by diglyceride acyltransferase (DGAT) and stored into cytosolic LD. Alternatively, FA might be incorporated in PL or CE after activation by acyl-coenzyme A synthetases (ACSSs) [41]. Consequently, de novo lipogenesis provides a diversity of FA employed in cell signaling and lipid homeostasis [24]. In particular, proliferating cancer cells undergo the overactivation of de novo lipogenesis to sustain their high need for lipids [42,43]. Moreover, the expression of FASN has been associated with cancer progression and drug resistance in preclinical models of breast cancer development [44,45]. Lipid synthesis dependent on FASN has been demonstrated to play a key role in metabolic adaptation to VEGF therapy in different tumor types [13,46]. Essentially, lipid synthesis by ACSS2 in the acetate pathway has been shown to support cancer cell survival under metabolic stress [47], suggesting an essential role of lipids in cancer cell adaptation to harsh environments. The nature of lipids forming plasma membrane can influence cell membrane fluidity and promote cancer cell movement and invasion. Phospholipids are major components of cell membranes and vary in chain length and saturation, two parameters that influence the fluidity and curvature of the membranes they compose. Phospholipids also play a role in homeostasis, cell adhesion, signal transduction, vesicle transport, apoptosis and posttranslational modifications. PL are mainly represented by phosphatidylcholine (PC), produced via the Kennedy pathway from PA [48] and PE produced by head group exchange from PS. Phospholipid composition is also maintained by the remodeling process Lands’ cycle [49]. Cholesterol is also present in all membranes; it controls the fluidity and flexibility of the membranes and plays an important role in the regulation of membrane function. Cholesterol is obtained from Ac-CoA via the mevalonate pathway (MVP) [50] or from extracellular LDL via the LDL receptor (LDLR) [12,51]. Finally, excess cholesterol and PL from peripheral tissues are transported to the liver in high-density lipoproteins (HDL) by the reverse cholesterol transport (RCT) process [19]. Evidence of LDL-induced breast cancer progression was revealed on human and mouse BC cells treated with LDL [52] and mice bearing breast tumors and fed with a high cholesterol diet [53]. The contribution of increased levels of LDL in breast cancer progression has been elegantly reviewed by Guan and colleagues [54]. However, investigations on the molecular mechanisms by which LDL regulates BC progression and the exploration of the clinical significance of LDL and therapy response are needed to clarify their impacts on BC patient outcomes.

#### 2.1.4. Lipid Catabolism

Lipid catabolism mainly includes TAG lipolysis (described above), and the β-oxidation of FA (FAO). Fatty acid oxidation is a cyclic process responsible for the translocation of long-chain Acyl-CoA to Ac-CoA across the mitochondrial membrane by the rate-limiting carnitine palmitoyl transferase-1 (CPT1). Fatty acid oxidation is mainly controlled by four enzymes: acyl-CoA dehydrogenase, enoyl-CoA hydratase, 3-hydroxyacyl-CoA dehydrogenase and ketoacyl-CoA transferase [55]. First, Acyl-CoA is converted in acylcarnitine by combination with carnitine catalyzed by CPT1, followed by translocation into the mitochondria via carnitine acyl carnitine translocase (CACT). Finally, acylcarnitine is converted back to Acyl-CoA by CPT2, and Acyl-CoA enters the FAO pathway. In each cycle, FAD-dependent dehydrogenation and NAD-dependent oxidation lead to the formation of FADH2 and NADH. The complete oxidation of the produced Ac-CoA, NADH and FADH2 is then accomplished by the TCA and oxidative phosphorylation [12,56]. In BC, a tumor specific variant of CPT1 (CPT1A) that lost its ability of fatty acyl transport to the mitochondria promotes survival, resistance to apoptosis and invasion by a mechanism dependent on the increasing activity of HDAC in BC cells [57]. The regulation of lipids can influence cell redox balance through biosynthesis and the remodeling of polyunsaturated FA (PUFA) in cell membranes. These steps require the enzymes ACSL4 [58] and lysophospholipid acyltransferase 3 (LPCAT3) (Figure 2). Briefly, ACSL4 catalyzes the combination of free arachidonic acid (AA) or adrenic acid (AdA) and CoA to form the derivatives AA-CoA or AdA-CoA, respectively, then LPCAT3 promotes their esterification to membrane PE to form AA-PE or AdA-PE [59]. An amount of malondialdehydes (MDA) is produced by AA-PE or AdA-PE oxidation and ultimately leads to ferroptosis. ACSL4 is considered the key enzyme to regulate lipid oxidative response and thus accelerates ferroptosis. The expression of ACSL4 is regulated by certain molecules, such as special protein 1 (Sp1), a transcription factor that upregulates ACSL4 transcription and promotes ferroptosis [60].

Therefore, lipid metabolism is regulated through diverse pathways, according to the cell type and the local context, thus giving cancer cells a controlled and adaptive supply of energy, membrane components and signaling messengers. Indeed, the vital role of lipids for cancer cell functions forces their metabolism to adapt rapidly to the harsh TME and to therapy pressure.

### 2.2. Factors of BC Plasticity and Lipid Metabolism Reprogramming

Breast cancer diversity may depend on the patient, cell of origin, stage of the disease, treatment and differences between the primary tumor and metastasis (Figure 1). Some of these differences result from genetic and epigenetic alterations, whereas others may reflect nonhereditary mechanisms such as tumor-specific adaptive responses. Intra-tumor and inter-tumor heterogeneities may simply help explain BC subtypes, as tumors composed of different “mixtures” of cancer cells [4]. This dynamic diversity, combined with lipid metabolism as complex system, multiplies BC phenotypes and cell behaviors.

#### 2.2.1. Intrinsic Factors of Heterogeneity

Oncogenesis driven by mutations in proto-oncogenes and tumor suppressor genes can affect downstream pathways such as cell metabolism [61]. Combined mutations can affect lipid metabolism via parallel events with additive effects or via a chain reaction due to linked pathways. For example, the modification of a metabolic enzyme through its expression, activity or half-life by a single or combined mutations can affect the concerned metabolic pathway. An inspiring review [62] has emphasized the fact that every tumor develops a unique metabolomic signature, coming from single or combined mutations, but this signature might be useful when pointed as a marker of therapeutic sensitivity.

Epigenetics drive the accessibility of transcription factors to genes via the regulation of methylation and the acetylation of histones and DNA. These regulations are driven by specific enzymes with opposite roles on methylation and acetylation [63]. Epigenetics determines cell specialization and function in normal tissues and promotes heterogeneity in tumors. Breast cancer cells have been shown to regulate their lipid metabolism through epigenetic adaptations. Similar to genetic mutations, epigenetic regulations may also have an important effect on cell metabolism that in turn can also play a role in epigenetics. Breast cancer cells enhance aerobic glycolysis via the Warburg effect to produce lactate and also use glutamine, folate and acetate to accelerate lipid biosynthesis. The glutamine-oriented metabolism protects from ROS elevation and apoptosis and now it is a potential and effective target for BC treatment. Moreover, because of their addiction to lipids for biomass, BC cells increase the de novo synthesis of FA instead of lipid uptake from the TME. This increase is allowed by higher expressions of several lipid metabolic genes, which are correlated with pathological features of BC, such as proliferation, metastasis and drug resistance [14,15,35]. The established link between metabolism and drug resistance is an attractive field of cancer research since the emergent concept that metabolism can play the role of a shield to drugs for tumor cells, shaping therapeutic resistance [56]. Redox balance can be dependent on lipid regulation pathways and modify the BC phenotype. For instance, the upregulation of LPCATs has been demonstrated to decrease the lipid oxidative response in BC [12,64]. Not only cholesterol but also oxysterols, metabolites from oxidized cholesterol, may play a role in BC progression and invasion. For example, oxysterols stimulate the proliferation and migration of BC cell lines and bone metastasis formation in BC patients [65]. Moreover, oxysterols can have a p53 inhibitory activity, thus promoting cell proliferation in an ER-dependent way in luminal BC [66].

Under normal conditions, the nature of the tissue and the type of cells have an inherent role on cell metabolism and homeostasis. Metabolic alterations in tumor cells are a consequence of new phenotypical needs for migration and invasion that rely on cancer cell capabilities to sustain energy and biomass through metabolism adaptation and plasticity [67]. Moreover, tumor cells often keep the metabolic behavior of the parental tissue through tissue-specific epigenetic regulations. This might imply that the same oncogenic driver may induce a different metabolic phenotype from one organ to another [68,69]. Furthermore, tumors originating from different cells within the same tissue may adopt different metabolic behaviors [62]. In addition, the tumor clonal composition and cellular phenotypes could change during tumor progression, which occurs in concert with metabolic changes at different stages of the disease [4].

##### Tumor Protein 53

Tumor protein 53 (p53) is a major tumor suppressor transcription factor, whose mutations lead to oncogenesis in many cancers. Lipid regulation represents a switch for BC and is often controlled by mutant p53, which accelerates lipid accumulation and could further contribute to cancer progression. Nevertheless, BC might use FAO to increase energy generation or drug resistance through the activation of signaling pathways, such as PI3K/AKT/mTOR JAK/STAT3 [56]. Otherwise, wild-type (WT) and mutant p53 in BC have generally opposite roles in lipid metabolism regulation. For example, mutant p53 depletion is sufficient to phenotypically revert BC cells to a more acinar-like morphology, thus dampening their disorganized morphology. The mevalonate pathway (MVP) was found upregulated in BC cells and identified as a lever for p53 mutant to increase cholesterol biosynthesis. Also, mutations in p53 can prevent the natural SREBP inhibitory activity of p53, leading to increased lipid metabolism via MVP in BC [70]. Moreover, a study reported that about 38% of aromatase inhibitor-resistant BC carry p53 mutations and they are associated with high-grade histology and high proliferation rates, corroborating the correlation between p53 mutation, lipid metabolism and tumor architecture [71]. 

Tumor protein 53 also plays a supervisor role in lipid metabolism by regulating gene expression or binding directly to metabolic enzymes. Tumor protein 53 generally tends to inhibit lipid synthesis, whereas mutant p53 enhances lipid anabolism. For example, p53 can inhibit the expression of SREBP1 [19], or inhibit glucose-6-phospate dehydrogenase (G6PD) activity by binding to it, leading to the inhibition of nicotinamide adenine dinucleotide phosphate (NADPH) production, an essential co-factor for de novo FA synthesis [72]. Moreover, p53 can suppress the MVP by upregulating the expression of ATP-binding cassette transporter (ABCA1) and limiting the activation of SREBP2 [73]. Tumor protein 53 also regulates FAO directly by controlling enzymes involved in the production of substrates essential for FAO. For instance, p53 activates pantothenate kinase-1 (PANK1) to enhance intracellular CoA content, a ubiquitous co-factor crucial for FAO, thus directly promoting FAO [19,74].

##### RAS

Evidence of RAS-induced lipid metabolic reprograming is well documented through several works demonstrating its role in pancreatic cancer through the control of lipid uptake, storage by the use of LD and through regulation of hormone-sensitive lipase (HSL) [75]. More recent studies have shown that cancer cells carrying mutant KRAS exert their tumorigenic effect through enhancing crosstalk within the TME, such as exchanging cytokines, growth factors and metabolites to improve metabolic adaptation to low nutrient availability [76,77]. In BC, the frequency of KRAS mutations is low, which has undermined its role in this disease, while RAS activity has been associated with TNBC progression [78]. The role of RAS in lipid metabolism, metastasis dissemination and drug resistance in TNBC might be dependent on other varied oncogenic alterations. Therefore, more investigations are still required to decipher the role of several genetic alterations that cooperate with KRAS mutations and lipid metabolism in BC.

Taken together, intrinsic factors can affect tumor cell metabolic heterogeneity and their adaptability to metabolic stress.

#### 2.2.2. Extrinsic Factors Regulating Lipid Metabolism in BC

The cell-extrinsic factors of metabolic heterogeneity include nutrient availability and interactions with the TME. Nutrient and oxygen availability might impact the pH surrounding the tumor cells, leading to glucose and lipid metabolic rewiring, changes in cellular composition within the TME and changes in blood flow and tumor pressure. Hydrostatic pressure applied to tumor cells has been shown to exert a selective expression of genes involved in metastasis and tolerance to oxidative stress; further investigations are needed to decipher if hydrostatic pressure can have a direct or an indirect role on metabolism [79]. Nevertheless, the study of the effects of physicochemical settings within the TME on tumor metabolism remains a tough task, mainly because of the technical difficulties of reproducing exact TME pressure conditions in vitro. Indeed, conventional culture media and even interstitial fluids from tumors differ from real TME conditions and conventional culture media cannot reproduce exhaustively the nutrient diversity in the TME that is rather dominated by the effects of the oncogenes on cell-intrinsic metabolic preferences. Moreover, the TME conditions may change temporally and spatially depending on the tumor progression [62]. Environmental factors such as special diet, obesity and chronic inflammation may also impact tumor progression and thus metabolic changes [80]. Metabolic differences between tumor cells and the cells of the TME often reflect a low response to treatment [81].

Cells within the TME compete and cooperate with one another through different signaling axis and nutrient uptake pathways [82]. The dynamic metabolic crosstalk involves immune cells, cancer-associated adipocytes (CAA), CAFs, blood vessels, mesenchymal cells, bone marrow-derived inflammatory cells and the ECM, which influence the hallmarks and fates of tumor cells [9]. Several recent reviews have highlighted a metabolic symbiosis of cancer cells and stroma cells as a key mechanism of drug resistance and response to hypoxia and metabolic stress [56,83,84]. Growing tumors might be in constant metabolic competition with cells in the TME, which exert a selective pressure on tumor cells by sequestrating nutrients, thus leading to a selection of only the strongest and the most adaptive cells that can change their metabolic preferences and survive [10,85]. The major role of the TME in cancer progression has oriented a cancer therapy switch from a “cancer-centric” model to a “TME-centric” one [82,86]. Nutrient deficiency within the TME leads to waste product accumulation and metabolic stress, especially in primary BC [87]. Metabolic stress can power tumor cells to adopt a migratory phenotype. For instance, in conditions of glutamine deficiency tumor cells use asparagine instead to sustain migration [88,89]. In addition, systemic therapy, by imposing a challenging environment, can force cancer cells to adapt their metabolic pathways to lipid metabolism [90].

In the TME, both tumor cells and immune cells keep different plasticity in a dynamic manner. For instance, tumor-associated macrophages (TAM), representing around 50% of some solid tumors, might regulate tumor cell metabolism while their own metabolism is influenced by metabolites produced by cancer cells, exhibiting a cell–cell cross-talk [91].

The cell–ECM interaction may also modify the metabolism of tumor cells that have detached from the matrix. Briefly, these cells manage ROS accumulation induced by their detachment by adapting their glucose metabolism, thus keeping them able to metastasize [62]. Moreover, mechanical support and paracrine cellular interactions between tumor cells and ECM are detrimental for metabolic reprogramming during mammary tumor development [2].

Altogether, intrinsic and extrinsic factors regulate tumor metabolism and metabolic heterogeneity in a highly dynamic and complex TME.

##### Cancer-Associated Adipocytes (CAA)

Adipocytes are the main stromal cells in the breast and are not terminally differentiated cells, which makes them impassive to the external environment. Indeed, besides differentiation, they might undergo various phenotypic and functional alterations that differentiate them from the mature adipocytes in many contexts, especially in BC where tumor-modified adipocytes have been named CAA [92]. During tumor progression, close interactions of CAA with BC cells make them express fewer differentiation markers and more pro-tumoral molecules, which could contribute to tumor cell aggressiveness. Cancer-associated adipocytes have increased catabolic processes that lead to the release of metabolites such as lactate, pyruvate and free FA [92]. The uptake of free FA in BC cells is mediated by several proteins, including CD36, FABPs and CPT1 that enhance tumor progression and aggressiveness [93]. Aggressive tumor traits are notably consecutive to an elevated production of ROS by BC cells [85,94]. In addition, FABP4 secreted by CAA actively transports FA to tumor cells and the elevated expression of the FA transport proteins such as CD36 and FABP5, leading to an amplified transport of FA to tumor cells and boosting their proliferation [95].

##### Cancer-Associated Fibroblasts (CAF)

Fibroblasts synthesize the ECM and are the most common cells of the connective tissue in animals. Due to the lack of specific markers to precisely identify fibroblasts, they are recognized by their morphology, tissue position and expression of leukocytes and epithelial and endothelial cell markers. Contrary to CAA, CAF may have originated from several types of cells and are activated during inflammation and fibrosis within the tumors [96]. Once activated, CAF interact with tumor cells continuously, promoting the proliferation and recruitment of each other and ultimately leading to tumor progression [97]. More specifically, CAF were shown to promote BC malignancy by secreting factors, generating exosomes, releasing nutrients, reshaping the ECM and suppressing the function of immune cells [68,98]. Cancer-associated fibroblasts influence the membrane fluidity of tumor cells and force them to adopt a higher invasive behavior. Mechanistically, CAF in contact with BC cells stimulate the expression of the key desaturase SCD1, which promotes the synthesis of MUFA and increases cell membrane fluidity and the migration properties of BC cells [99].

##### Tumor-Associated Macrophages (TAM)

Tumor-associated macrophages exert ambiguous behaviors in tumor development and progression, depending on the type of cancer they are associated with, but they always play a critical role. A study reported that the expression of adipocyte fatty acid-binding protein (A-FABP) in TAM promotes BC progression [100]. Moreover, a new type of lipid-associated macrophage has been found in BC, highly expressing FABP5 but not in the conventional M1/M2 classification. These TAM also express programmed-death ligand 1 (PD-L1) and PD-L2 and exert an anti-tumoral response. Considering the rich lipid sources of breast TME and the dependence on FA of tumor cells, lipid-associated macrophages may belong to the tissue-resident macrophage population that is reprogrammed by tumor cells [101,102]. On the contrary, another study showed that epidermal fatty acid-binding protein (E-FABP) is highly expressed in macrophages, particularly in a specific subset, promoting their antitumor activity. In tumor stroma, E-FABP-expressing TAM produce high levels of IFN-β through the upregulation of LD formation in response to tumors and enhance the recruitment of natural killer cells (NK) [103]. Fatty acid-binding proteins might have distinct roles in lipid metabolism within the TME; more research is needed to decipher their role when expressed by TAM.

##### Tumor-Infiltrating Lymphocytes (TIL)

Tumor-infiltrating lymphocytes include innate lymphoid cell T and B cells. Breast cancer is traditionally considered as poorly immunogenic, but the TNBC and HER2-positive subtypes show a high level of TIL, indicating that an immunotherapeutic approach may be suitable for this hard-to-treat malignancy [104,105]. In CD8 TIL, SREBP2 signaling is essential for their proliferation and effector function. Increased cholesterol synthesis and uptake by TIL enhances their antitumor effect against BC. Moreover, the inhibition of Acetyl-Coenzyme A acetyltransferase 1 (ACAT1) in CD8 TIL alters the synthesis of cholesterol and leads to an accumulation of free cholesterol in the plasma membrane, which binds directly to the T cell receptors (TCR). The T cell receptors are then clustered, mimicking antigen-induced signals and in turn increasing cholesterol biosynthesis and uptake. Furthermore, this cholesterol helps in the formation of mature immunological synapses for the targeted killing of tumor cells. On the other hand, cholesterol accumulation in the TME induces ER stress and increases T cell exhaustion. Thus, the functions of endogenous and exogenous cholesterol may differ [12].

## 3. Health and Physiologic Consequences

The challenge of transforming scientific data from laboratory experiments to efficient treatments is an inevitable brake to the development of new treatments, especially in oncology. Indeed, all cancer models partially or imperfectly reproduce patient tumors. This is inherent to the use and study of all models, resulting in an important rate of clinical trial failures for drugs that were however promising in previous steps. Advances in technologies such as -omics and high-throughput analysis systems now allow us to analyze tumors in unprecedented depth, reducing the difficulties linked to the diversity of cancer cell populations and their microenvironment. Combining immune and metabolic pathways, an increasing arsenal is now pushing the limits of cancer treatment.

### 3.1. Hypoxia

The link between hypoxia and lipid metabolism in cancer is now evident. Under the hypoxic and metabolic stress conditions encountered during cancer development, there are multiple transcription factors and enzymes that are activated in cancer cells to cope with cell stress and the nutrient and energy needs for survival in a harsh TME. Hypoxic signaling involves several mechanisms that directly contribute to the deregulation of glycolysis, acidosis, lipid metabolism, ECM remodeling and ultimately BC aggressiveness [106]. The crosstalk between hypoxia and the ECM in orchestrating metabolic alterations through YAP/TAZ signaling has been demonstrated to play a role in cancer progression [107]. Mechanisms regulating lipid metabolism under hypoxia involve mainly the expression of the central key regulators of FA metabolism, SREBP1 and FASN, by hypoxia-inducible factors (HIFs). Hypoxia-inducible factor signaling activates FA synthesis by inhibiting CPT1, responsible for FA transport into the mitochondria and the storage of FA in LD. Moreover, HIF expression is upregulated by ER stress in order to decrease cytotoxic ER stress by forming LD, and the repression of SREBP1 or limitation of FASN can also activate the HIF-1a signaling pathway and the unfolded protein response [108]. In the context of energy deficiency-mediated stress, HIF signaling pathways coordinate with 5′ adenosine monophosphate-activated protein kinase (AMPK) and the mammalian target of rapamycin (mTOR) to compensate for the limitation of FASN and activate lipid metabolism to rescue lipid-mediated ER stress [24].

Gene copy number multiplication is also a genetic adaptation that gives metabolic advantages to BC in certain contexts. For example, BC exhibits an increased number of copies coding for the enzyme ACSS2, making them more resistant to hypoxia and nutrient deprivation. A positive correlation between ACSS2 expression and tumor progression has even been revealed [10,47].

### 3.2. Lipid Peroxidation

Redox equilibrium is compulsory for all biological systems, balancing oxidative and reducing reactions to achieve suitable conditions for life. An accumulation of oxidizing molecules either by the overproduction or loss of cellular reducing ability leads to the oxidation of DNA, proteins, and lipids, thereby altering their structure, activity, and physical properties. Elevated ROS leads to the oxidation of PUFA and generation of reactive aldehydes, such as 4-hydroxyalkenals, malondialdehyde and acrolein, which have been involved in many diseases, including cardiovascular, inflammatory, and metabolic diseases. These aldehydes have damaging effects on cell membrane lipids by generating phosphatidylethanolamine (PE)-derived adducts and may affect the membrane property and the functions of membrane transporters, channels, receptors and enzymes [109]. The excessive oxidation of lipids alters the physical properties of cellular membranes and can cause the covalent modification of proteins and nucleic acids [110]. Peroxidation is an oxidation reaction that produces peroxide. Enzymatic peroxidation is mostly mediated by lipoxygenases (LOX) that catalyze the stereospecific insertion of oxygen into polyunsaturated fatty acids (PUFA), especially arachidonic acid (AA) and adrenic acid (AdA), the most susceptible to peroxidation [43]. Lipid peroxides are direct inducers of a recently discovered non-apoptotic cell death: ferroptosis [111]. Since the important role of lipid ROS in ferroptosis was revealed, there has been much interest in understanding which lipid species are involved in the regulation of ferroptotic cell death. Upon ferroptosis induction, most oxygenated PL species are upregulated, suggesting that ferroptosis ultimately damages most membrane PL [59,112].

### 3.3. Drug Resistance and Clinical Outcomes

Because of their important role in cell growth and signaling, lipids might influence the drug resistance of tumor cells. Based on preclinical models, lipid metabolism inhibition managed to dampen the drug resistance of cancer cells. These successes rely mainly on the capacity of a remodeled lipid metabolism to limit the stress induced by anticancer treatments and by counteracting oxidative and metabolic stresses [113]. Lipid metabolism is governed by enzymes and transcription factors through functional studies evidenced by targeting these regulatory proteins using a panel of methods and technologies: siRNA, blocking or neutralizing antibodies, small inhibitory molecules, genetic KO, etc. In order to exacerbate the expected effects, these inhibitors can be combined with more classical cancer therapies such as chemotherapy and immunotherapy. Moreover, reprogrammed lipid metabolism in cancer cells passes mainly through the activation of FAO, increased FA de novo synthesis, the aberrant accumulation of LD and changes in the lipid composition of cell membranes [25,56]. Thus, targeting key enzymes in these steps might abolish the drug resistance of tumors cells. Recent inspiring reviews have already listed and described the major targeted enzymes to treat drug resistance in BC (Table 1). The positivity of BC markers might be strongly influenced by metabolic enzymes. That is the case with FASN, whose activity is positively correlated with the positivity of the HER2 marker, cancer progression and chemoresistance. Furthermore, the overexpression of FASN is remarkably associated with relapse and metastasis in patients with HER2-enriched BC [35,114]. Cholesterol metabolism is also a weapon used by BC, mainly via the reprogramming of cholesterol de novo synthesis. This reprogramming is permitted by an upregulation of SREBP2, which activates MVP and thus produces cholesterol from Ac-CoA [12,115].

The overexpression of lipid metabolism genes is one of the numbered genetic adaptations of BC. For example, the overexpression of ACSL4 is associated with the invasiveness of BC [117]. On the contrary, the suppression of SREBP1 significantly inhibited the migration and invasion of BC cell lines, even constituting a prognostic marker [118]. Moreover, the overexpression of ACLY increases lipogenesis and suppresses cell senescence at the same time by a higher activity of ACLY and a lower p53 expression [35,119]. Besides, a mutation on the p53 gene has been revealed to have an impact on the expression of several lipogenic genes. The p53R273 mutant increases the expression of FASN, ELOVL6 and SCD1, leading to an increased FA de novo synthesis via the MVP and tumor progression [70].

A higher lipid metabolism in BC is also linked to the organ in which metastasis settles, such as the brain. Ferraro et al. discovered and explained this phenotype as an adaptation to decreased lipid availability in the brain compared with other tissues, resulting in site-specific dependency on FA synthesis for breast tumors growing in the brain. The inhibition of FASN reduced HER2-positive breast tumor growth in the brain [24,120]. Lipid metabolism may also help to diagnose and identify BC with precision. Indeed, two studies showed that free FA, PL and metabolite distribution could allow the deciphering of BC tissues from healthy tissues with high accuracy, ranging from 98% to 84% [121,122]. Moreover, increased levels of serum lipids and lipoproteins have been associated with BC risk. Further studies are needed to cluster the importance of factors including cancer stages, types of cancer, parity and menopausal status that may affect lipid profiles in BC [123]. The FA uptake is also important for BC progression, which has been demonstrated by the inhibition of CD36 that causes the reduced growth and viability of BC cells [124,125].

Gene expression signatures have delineated clinically distinct subtypes of BC [126], and these subtypes exhibit metabolic differences defined by lineage-specific gene expression patterns. For example, glutamine synthesis and glutamine consumption change from basal subtypes to TNBC, notably because of the variable expression of glutamine metabolism enzymes [62]. Indeed, TNCB show a bigger need for glucose and glutamine, and require more exogenous lipid uptake and storage [127]. It was also shown that within BC subtypes, transcriptome is associated to the lipidome [128]. For instance, CPT1A was found upregulated in hormone receptor-positive BC and plays a key role in cell proliferation and drug resistance in this tumor type [55].

## 4. Conclusions

Cell metabolic plasticity represents the ability of cancer cells to rapidly reprogram their gene expression repertoire, to change their metabolic preferences and behavior and to adapt to microenvironmental nods. These characteristics also directly contribute to tumor heterogeneity and are critical for tumor malignancy. Metabolic heterogeneity is one of the most characteristic features of BC, which meets the enormous energy demands of tumor cell growth. Certainly, the lipogenic phenotype facilitates the malignant conversion in BC and in many other cancers. The addiction to lipids, and dependency on lipid synthesis, uptake, storage and oxidation were identified to support proliferation, energetics in the diverse context of cancer progression and therapy resistance. The interplay between intrinsic and extrinsic factors leads to the diverse activation of lipid metabolic pathways, which is considered as a key metabolic adaptation to the TME and to therapy stress. It also has a balanced equilibrium for the anabolic and catabolic pathways for cancer progression. The field of lipid metabolism in drug resistance in gaining interest, with a panel of key potential targets for the treatment of resistant BC. Therefore, tracking metabolic heterogeneity in BC might provide key hints for targeting relapse and drug resistance.

## Figures and Tables

**Figure 1 cancers-14-06267-f001:**
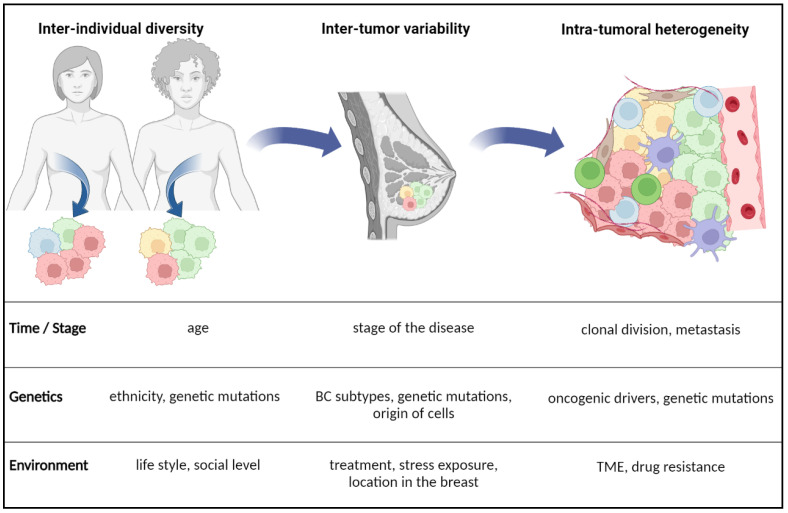
Different scales and factors of heterogeneity in breast cancer. Heterogeneity in breast cancer might be dependent on individuals, the environment and cell types within the tumor microenvironment. For each scale, main factors of heterogeneity (time, genetics and environment) may vary in their impacts and approach. For example, a single genetic mutation can be common to two unrelated women but differentially expressed within their respective tumors.

**Figure 2 cancers-14-06267-f002:**
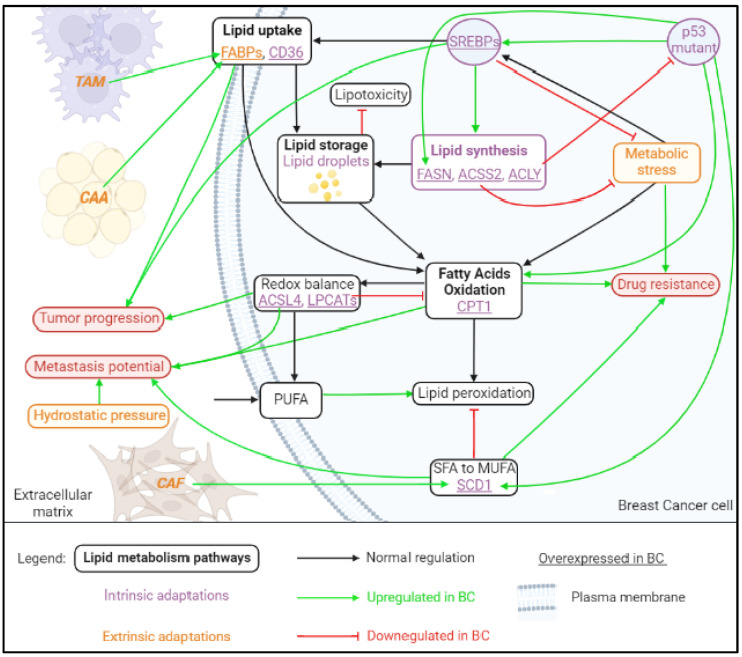
Known lipid metabolism adaptations in BC cells. Breast cancer cells can develop and undergo adaptation of their diverse lipid metabolic pathways to cover their high needs for energy and biomass synthesis. Main proteins responsible for these metabolic changes are depicted here and are deciphered as intrinsic proteins (in purple) or derived from the TME (in orange). Major effects of lipid metabolic pathways on BC progression are indicated by green arrows. Metabolic stress includes nutrient deprivation, hypoxia and endoplasmic reticulum stress. Upregulated and downregulated pathways described above illustrate an inherent large heterogeneity of lipid metabolism in BC. ACLY: ATP-citrate lyase, ACSL4: long-chain acyl-coenzyme A synthetase 4, ACSS2: acyl-coenzyme A synthetase 2, CAA: cancer-associated adipocytes, CAF: cancer-associated fibroblasts, BC: breast cancer, CD36: fatty acid-translocase, CPT1: carnitine palmitoyltransferase 1, FAPBs: fatty acid-binding proteins, FASN: fatty acid synthase, LPCATs: lysophospholipid acyltransferases, MUFA: monounsaturated fatty acids, p53: tumor protein 53, PUFA: polyunsaturated fatty acids, SCD1: stearoyl-CoA desaturase 1, SFA: saturated fatty acids.

**Table 1 cancers-14-06267-t001:** List of lipid enzymes and fatty acid transport proteins that are targeted by single agents or in combination with therapeutics for the treatment of BC.

Targets	References
FASN, Lipin, CD36, CPT1/2, SCD1	[56]
FASN, CD36, CPT1/2, SREBP1	[25]
FASN, ACLY, ACC, SREBP1, SCD1	[12]
FASN, CD36, ACLY, CPT1, ACC, ACS, SREBP1, SCD1	[24]
FASN, ACLY, ACSS, SCD1	[116]

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
