# Peer review of "Lipid Metabolism Heterogeneity and Crosstalk with Mitochondria Functions Drive Breast Cancer Progression and Drug Resistance"

_cancers, 2022, doi:10.3390/cancers14246267_

Round 1

Reviewer 1 Report

file attached

Author Response

Point by point answering letter.

Reviewer 1:

The review by Aurelien Azam and Nor Eddine Sounni addresses very important aspects of research concerning lipid metabolism heterogeneity and lipid metabolic pathways in breast tumorigenesis and therapy resistance. The review is very informative and is supported by two figures and one table. I suggest some changes/additions before the paper could be published.

Thanks to the reviewer 1 for the positive feedback on the review and for the proposed suggestions and corrections that helped us for improving this review.

We have addressed the comments as follows:

  1. 1 is confusing, add one more column on the left side and shift “Stage”, “Time”, “Genetics” and “Environment” in this column.

As suggested, we have changed the Figure1 by adding a column on the left side and make it more clear for the reader.

  1. 4 Please cite Kamp et al. (doi: 10.1021/bi00037a034) after ”Although FA can diffuse across phospholipid (PL) bilayers ..”

The reference by Kamp et al. was added as indicated in the revised manuscript.

  1. Chapters 3.1 (Hypoxia) and 3.2 (Lipid peroxidation) are somewhat superficial and are not supported by recent literature. Authors may consider to cite the recent reviews DOI: 10.3390/genes13091585 (Crosstalk between Hypoxia and Extracellular Matrix in the Tumor Microenvironment in Breast Cancer) and doi:10.3390/molecules24244545 (The Role of Phosphatidylethanolamine Adducts in Modification of the Activity of Membrane Proteins under Oxidative Stress).

As suggested by the reviewer the chapters 3.1 and 3.2 were revised by introducing several new works in the field and added the proposed references.

See changes from lines 633-638 in Chapter 3.1 for hypoxia and lines 702-707 in the chapter 3.2 related to lipid peroxidation.

  1. The entire text should be checked for grammar. Below are only few examples for the awkward or incorrect sentences.

We apologize to the reviewer for these mistakes and thanks for raising these errors that we have omitted. We have now corrected the manuscript as proposed and verified the whole text for missing grammatical and typing errors as indicated in the revised manuscript with tracked changes.

p.1 Metabolic heterogeneity of BC increases it ability to adapt tumor microenvironment changes and metabolic stress ...→ Metabolic heterogeneity of BC enhances its ability to adapt to changes in the tumor microenvironment and metabolic stress ...

p.2 line 67 Metabolic adaption → Metabolic adaptation

p.2 line 72 targeting it vulnerability and harnessing malignancy

p.4 Once into the cytosol → Once in the cytosol

p.4 entry to mitochondria through carnitine palmitoyltransferase 1 (CPTl) and fatty acid oxidation (FAO) → enter to mitochondria through carnitine palmitoyltransferase 1 (CPTl) and undergo fatty acid oxidation (FAO)?

p.4 LD consist in organelle?

Yes this has been specified as storage organelle and added a new supporting new reference (27) doi: 10.1038/s41580-018-0085-z) (See line 192 of revised manuscript with tracked changes version)

p.5 In particular, to support their high-lipid consuming metabolism, cancer cells aberrantly use this pathway to obtain FA and independently of exogenous uptake?

For clarity, this sentence has been changed. See line 279 of revised manuscript with tracked changes version  “In particular, proliferating cancer cells undergo overactivation of de novo lipogenesis to sustain their high-need for lipids [42,43] »

p.5 Finally, Excess cholesterol → Finally, excess of cholesterol

yes

p.5 previously described above → described above

yes

p.5 This sentence should be re-written to gain clarity: “In breast cancer, expression of a tumor specific variant CPTlA that does not retain the transferase activity of CPTl has been found to play a role in breast cancer cells survival, resistance to apoptosis and invasion through a non-FAQ mechanism through increased HDAC activity”.

For clarity, this sentence has been re-edited. See line 340 of revised manuscript with tracked changes version: “In breast cancer, a tumor specific variant of CPT1 (CPT1A) that lost its ability of fatty acyl transport to mitochondria promotes survival, resistance to apoptosis and invasion by a mechanism dependent on increasing activity of HDAC in BC cells [57]. »

p.6 drugs for tumors cells → drugs for tumor cells

p.6 Redox balance can dependents on lipid regulation pathways → Redox balance can depend on lipid regulation pathways

p.10 leads to oxidize DNA, proteins, and lipids → leads to oxidation of DNA, proteins, and lipids

These errors were corrected in the revised version (see revised manuscript with tracked changes version).

Reviewer 2 Report

The Review article “Lipid metabolism heterogeneity and crosstalk with mitochondria functions drive breast cancer progression and drug resistance” by Azam and Sounni summarized the lipid metabolism’ heterogeneity and its reprogramming to adapt to tumor microenvironment. The lipid metabolism in breast cancer cells is linked to its disease progression and drug resistance therefore providing new opportunities for therapeutic intervention. Overall the paper is well written except for a few grammar mistakes that I think the authors need to pay attention. I have a few suggestions listed below.

1.     There are many abbreviations in this paper, I suggest the author add a glossary of abbreviations to make the paper easier to read.

2.     The role of LDL (low density lipoprotein) in lipids transport/deliver and storage in breast cancer cells need to be discussed/added in the paper.

3.     Grammar mistakes: line 17, its ability; line 272, originated; line 278, numbers of; line 463-467, try to break the sentence apart; line 541-542, no punctuation mark in this sentence.  

Author Response

Point by point answering letter.

Reviewer 2:

The Review article “Lipid metabolism heterogeneity and crosstalk with mitochondria functions drive breast cancer progression and drug resistance” by Azam and Sounni summarized the lipid metabolism’ heterogeneity and its reprogramming to adapt to tumor microenvironment. The lipid metabolism in breast cancer cells is linked to its disease progression and drug resistance therefore providing new opportunities for therapeutic intervention. Overall the paper is well written except for a few grammar mistakes that I think the authors need to pay attention. I have a few suggestions listed below.

Thanks to the reviewer 2 for the positive feedback and for the constructive remarks that helped us to improve the manuscript.

We have addressed the comments as follows:

  1. There are many abbreviations in this paper, I suggest the author add a glossary of abbreviations to make the paper easier to read.

As suggested, we have added a glossary of abbreviation (see lines 812-834 of revised manuscript with tracked changes version)

  1. The role of LDL (low density lipoprotein) in lipids transport/deliver and storage in breast cancer cells need to be discussed/added in the paper.

We have added the role of LDL in breast cancer progression in the revised manuscript (see lines 300-306 of revised manuscript with tracked changes version)

  1. Grammar mistakes: line 17, its ability; line 272, originated; line 278, numbers of; line 463-467, try to break the sentence apart; line 541-542, no punctuation mark in this sentence. 

Thanks to the reviewer for raining these mistakes, we have corrected these errors and verified the whole paper for mislabeling and grammar mistakes (see revised manuscript with tracked changes version).